# Semaphorin 3E deficiency dysregulates dendritic cell functions: *In vitro* and *in vivo* evidence

**Hesam Movassagh**[1], **Lianyu Shan**[1], **Latifa Koussih**[1,2], **Abdulaziz Alamri**[1], **Nazila Ariaee**[1], **Sam K. P. Kung**[1], **Abdelilah S. Gounni**[ORCID][1]*

**1** Department of Immunology, Rady Faculty of Health Sciences, Max Rady College of Medicine, University of Manitoba, Winnipeg, Manitoba, Canada, **2** Department des Sciences Experimentales, Universite de Saint-Boniface, Winnipeg, Manitoba, Canada

* abdel.gounni@umanitoba.ca

**Data Availability Statement:** All relevant data are within the manuscript and its Supporting Information files.

## Abstract

Regulation of dendritic cell functions is a complex process in which several mediators play diverse roles as a network in a context-dependent manner. The precise mechanisms underlying dendritic cell functions have remained to be addressed. Semaphorins play crucial roles in regulation of various cell functions. We previously revealed that Semaphorin 3E (Sema3E) contributes to regulation of allergen-induced airway pathology partly mediated by controlling recruitment of conventional dendritic cell subsets *in vivo*, though the underlying mechanism remained elusive. In this study, we investigate the potential regulatory role of Sema3E in dendritic cells. We demonstrated that bone marrow-derived dendritic cells differentiated from $Sema3e^{-/-}$ progenitors have an enhanced migration capacity both at the baseline and in response to CCL21. The enhanced migration ability of Sema3E dendritic cells was associated with an overexpression of the chemokine receptor (CCR7), elevated Rac1 GTPase activity and F-actin polymerization. Using a mouse model of allergic airway sensitization, we observed that genetic deletion of Sema3E leads to a time dependent upregulation of CCR7 on CD11b⁺ conventional dendritic cells in the lungs and mediastinal lymph nodes. Furthermore, aeroallergen sensitization of $Sema3e^{-/-}$ mice lead to an enhanced expression of PD-L2 and IRF-4 as well as enhanced allergen uptake in pulmonary CD11b⁺ DC, compared to wild type littermates. Collectively, these data suggest that Sema3E implicates in regulation of dendritic cell functions which could be considered a basis for novel immunotherapeutic strategies for the diseases associated with defective dendritic cells in the future.

## Introduction

Dendritic cells (DC) are key inflammatory cells bridging innate to adaptive immune response. Function of dendritic cells is a key determinant to shape immunity in several inflammatory conditions. Therefore, addressing the mechanisms and mediators involved in regulation of dendritic cell functions has a tremendous impact on understanding how the immune response

**Funding:** This work was supported by the Canadian Institutes of Health Research grant (PJT # 173291). H. M. was supported by Research Manitoba-Children's Hospital Research Institute of Manitoba Studentship. The funders had no role in study design, data collection and analysis, decision to publish, or preparation of the manuscript.

**Competing interests:** The authors have declared that no competing interests exist.

**Abbreviations:** DC, dendritic cells; FACS, fluorescence-activated cell sorting; HDM, house dust mite; IL, interleukin; I.N., intranasal; mAb, monoclonal antibody; MLN, mediastinal lymph nodes; SEM, standard error of mean; Sema, semaphorin.

could be initiated, developed, and impaired. Furthermore, it may introduce novel targets to design innovative immunotherapeutic strategies [1]. In the context of allergic asthma, programmed death ligand B7DC (PD-L2) has been shown to be upregulated after allergen sensitization in murine myeloid DCs which correlates with the severity of the disease in humans. Allergen-induced upregulation of PD-L2 decreases IL-12 level and eventually aggravates airway hyperresponsiveness (AHR); while PD-L2 blockade dampens AHR development [2]. In addition, it has been previously shown that CD11b$^+$ conventional DC (cDC) play a pivotal role in induction of type 2 inflammation via induction of CCR7-mediated migration from the lungs to the lymph nodes and uptake of aeroallergens such as house dust mite (HDM) [3]. This process is tightly regulated by a transcription factor, interferon regulatory factor 4 (IRF-4) [4, 5]. Therefore, it is essential to decipher the novel mediators involved in regulation of CCR7, IRF-4, and PD-L2 in allergic asthma.

Semaphorins, originally identified in the nervous system, are a versatile family of guidance cues which are ubiquitously expressed and function in different organ systems including the immune system [6]. Immune semaphorins, categorized as class IV transmembrane molecules, control essential cell functions such as migration and proliferation as well as cytokine and antibody response. A compelling body of evidence suggests that other semaphorins such as those of secreted class III ones could play a key role on development of immune cells and also orchestration of their functions after development [7]. Semaphorin 3E (Sema3E) and its receptor, PlexinD1, have been previously shown to be involved in development of T cells by regulation of their chemokine-mediated migration from cortex to medulla of the thymus [8]. PlexinD1 has also emerged as a negative regulator of IL-12/IL-23p40 production in DC [9]. We have previously reported an essential regulatory role of Sema3E in mouse model of allergic asthma which is mediated in part through modulating DC functions *in vivo* [10]. However, the precise mechanism underlying this role is not clear.

Here, we address the non-redundant role of Sema3E in allergen sensitization, DC migration, and antigen uptake. We further demonstrate that the role of Sema3E in CD11b$^+$ cDC functions is mediated by regulation of CCR7, IRF-4, and PD-L2 expression. Finally, F-actin polymerization and Rac1 GTPase activity is enhanced in DC differentiated from *Sema3e$^{-/-}$* mice *in vitro*. This may be relevant to dysregulated immune response in which the exacerbated DC function takes the center stage.

## Materials and methods

### Animals

The 129 P2 *Sema3e$^{-/-}$* mouse was a kind gift from Dr. F. Mann (Developmental Biology Institute of Marseille Luminy, Université de la Méditerranée, Marseille, France) which was described previously [11] and 129P2 WT littermates were used as control groups. All the mice were maintained at the Central Animal Care Services (CACS) facility at the University of Manitoba under specific pathogen-free conditions and used according to guidelines stipulated by the Canadian Council for Animal Care and approved by the University of Manitoba Animal Care and Use Committee (Protocol Number 15802).

### Differentiation of bone marrow-derived dendritic cells (BMDCs)

Naive 129P2 WT and *Sema3e$^{-/-}$* mice were euthanized, their femurs and tibias were dissected, and the BM was flushed out by injecting complete DMEM through the marrow cavities and cells were cultured for 7 days in DMEM containing mouse recombinant GM-CSF (PeproTech, Rocky Hill, NJ) followed by LPS-induced DC maturation. The preparations were stained with

CD11c antibody followed by FACS analysis to assess their purity [12]. In some experiments, surface expression of CCR7 was studied on BMDC DC by using FACS analysis.

**BMDC migration assay.** BMDCs from *Sema3e*$^{-/-}$ or WT mice were seeded ($15 \times 10^4$/well in 0.1 ml) in the upper compartment of a transwell chamber. 20 ng/ml of mouse recombinant CCL21 (PeproTech, Rocky Hill, NJ), or PBS as vehicle, was added to the lower compartment as a chemoattractant. After 4h incubation at 37˚C, migrated cells towards the lower chamber were counted and compared with control groups.

## Rac1 GTPase activity

BMDCs from *Sema3e*$^{-/-}$ and WT mice were first stimulated with CCL21 (20 ng/ml) for 0, 0.5, 1, 5, 15 and 30 min. Then, GTPase activity of Rac1 was measured in snap-frozen BMDC extracts by G-LISA activation assay according to the manufacturer's instructions (Cytoskeleton, Denver, CO).

## Actin polymerization

BMDCs were stimulated with CCL21 (20 ng/ml) for 0, 0.5, 1, and 5min and immediately fixed with 4% paraformaldehyde. Then, cells were washed and permeabilized with %0.05 Triton-X100 in for 30 min before F-actin staining with Alexa488 conjugated Phalloidin (Life Technologies) for 30 min. Finally, the CD11c$^+$ pre-gated BMDCs were analyzed by flow cytometry to detect F-actin content.

## Aeroallergen sensitization model

Lyophilized HDM protein extract (*Dermatophagoides pteronyssinus*, Lot 259585; LPS, 615 EU/vial was obtained from Greer Laboratories (Lenoir, NC) which was reconstituted in sterile saline as 2.5 mg/ml stock concentration before treatment. A single intranasal dose (100 **μ**g in 100 **μ**L of saline) was freshly administered to *Sema3e*$^{-/-}$ or WT mice under gaseous anesthesia [13].

## Flow cytometric analysis of pulmonary dendritic cells

Pulmonary conventional DC subsets were analyzed by FACS from *Sema3e*$^{-/-}$ or WT mice 3 days after intranasal exposure with a single high dose of HDM [3]. Briefly, lungs were removed from mice and enzymatically digested using 1 mg/ml collagenase IV (Worthington Biochemical Corporation, Lakewood, NJ) and 0.5 mg/ml DNase from bovine pancreas in RPMI 1640 medium. After Fc blocking, DCs were stained by anti-mouse CD11c-APC (Clone: N418, eBioscience), MHCII eFluor® 450 (Clone: M5/114.15.2, eBioscience), CD11b-PE-Cy7 (Clone: M1/70, BioLegend), and CD103-PerCP-Cy5.5 (Clone: 2E7, BioLegend). Anti-mouse PD-L2-PE (Clone: TY25, BioLegend) and IRF-4 (Clone: IRF4.3E4, BioLegend) antibodies were separately added to the tubes followed by acquisition of the samples using a BD FACS Canto-II (BD, San Diego, CA) and analyzed using FlowJo V10.7.

## Labeling HDM and *in vivo* uptake assay

HDM was labeled with the Alexa Fluor 647 protein Labeling Kit (Molecular probes, Life Technologies Inc.) and purified with resin column following manufacturer's instructions. Mice were sensitized intranasal with 100 **μ**g AlexaFluor 647- HDM or non-labeled HDM for 36 hours. Lung single cells suspension was purified as described above [10], then stained using V.D-Fixable Viability Dye eFluor™, 780/dumping-PE(CD4,CD8, 6G(1A8), Siglc-F, B220,

MHCII-PB, CD11c-FITC, CD103-PerCy5.5, CD11b-PE-Cy7, HDM-APC antibodies. Samples were then using a BD FACS Canto-II (BD, San Diego, CA) and analyzed using FlowJo V10.7.

## Statistics

GraphPad Prism 5.0 software was used for statistical analysis and values were presented as the mean±SEM of at least three independent experiments. Depends on the number of groups and treatments, data were analyzed by unpaired $t$ test, one-way or two-way ANOVA, followed by the Bonferroni's multiple comparison post-hoc test. Differences were considered to be statistically significant at $^*p{\leq}0.05$, $^{**}p{\leq}0.01$ and $^{***}p{\leq}0.001$.

## Results

### Sema3E deficient BMDC display an enhanced migratory phenotype *in vitro*

To support our previous *in vivo* findings on the hyper migratory phenotype of *Sema3e*$^{-/-}$pulmonary DC, we first differentiated progenitors harvested from the bone marrow to DC in the presence of GM-CSF *in vitro* [12].This approach not only enables us to investigate the functional outcomes reproducibly but also furthers the signaling mechanistic underlying DC dysfunction in the absence of Sema3E. Prior to functional studies, we assured the purity of *in vitro* differentiated BMDC by FACS analysis of CD11c surface expression (>95%, data not shown). Interestingly, BMDC from *Sema3e*$^{-/-}$ mice expressed a higher level of this marker compared to WT BMDC (S1 Fig). Transwell migration experiments revealed that BMDC from *Sema3e*$^{-/-}$mice had higher basal migration compared to the BMDC from WT littermates. Stimulation of BMDC with CCR7 ligand, CCL21, induced higher migration in Sema3E-deficient BMDC than the WT controls (Fig 1A).

To address the mechanism underlying the role of Sema3E in the regulation of BMDC migration we next examined the surface expression of CCR7 which specifically binds CCL21. As depicted in Fig 1B, basal and CCL21-induced surface expression of CCR7 was higher in BMDC from *Sema3e*$^{-/-}$mice than those of WT counterparts. In addition, stimulation with an aeroallergen, HDM upregulated surface expression of CCR7 in the absence of Sema3E (S2 Fig). Therefore, Sema3E may exert its inhibitory effect on DC migration at least partly through regulation of CCR7 expression at both homeostatic and allergic inflammatory conditions.

### Genetic ablation of Sema3E is associated with increased Rac1 GTPase activity and F-actin polymerization

Activation of small GTPases particularly Rac1 is indispensable for the acquisition of the migratory cell phenotype [14]. Hence, we compared the level of Rac1 GTPase activity (GTP-bound Rac1) between BMDC from *Sema3e*$^{-/-}$and WT mice by performing luminescence-based G-LISA. Our studies revealed that Rac1 GTPase activity is significantly higher in the absence of Sema3E one minute after CCL21 stimulation as an early signaling event (Fig 1C). Furthermore, staining of filamentous actin (F-actin) by Phalloidin assay demonstrated that actin polymerization, as an essential component of cell migration [15], was remarkably enhanced in BMDC from *Sema3e*$^{-/-}$compared to the WT mice upon CCL21 stimulation at indicated time points but not at the baseline (Fig 1D). Therefore, the regulatory role of Sema3E in BMDC functions could be mediated by its effect on Rac1 GTPase and actin rearrangement. Altogether, our mechanistic data suggest that in the absence of Sema3E, two key signaling events including Rac1 GTPase activity and F-actin assembly are dysregulated in dendritic cells.

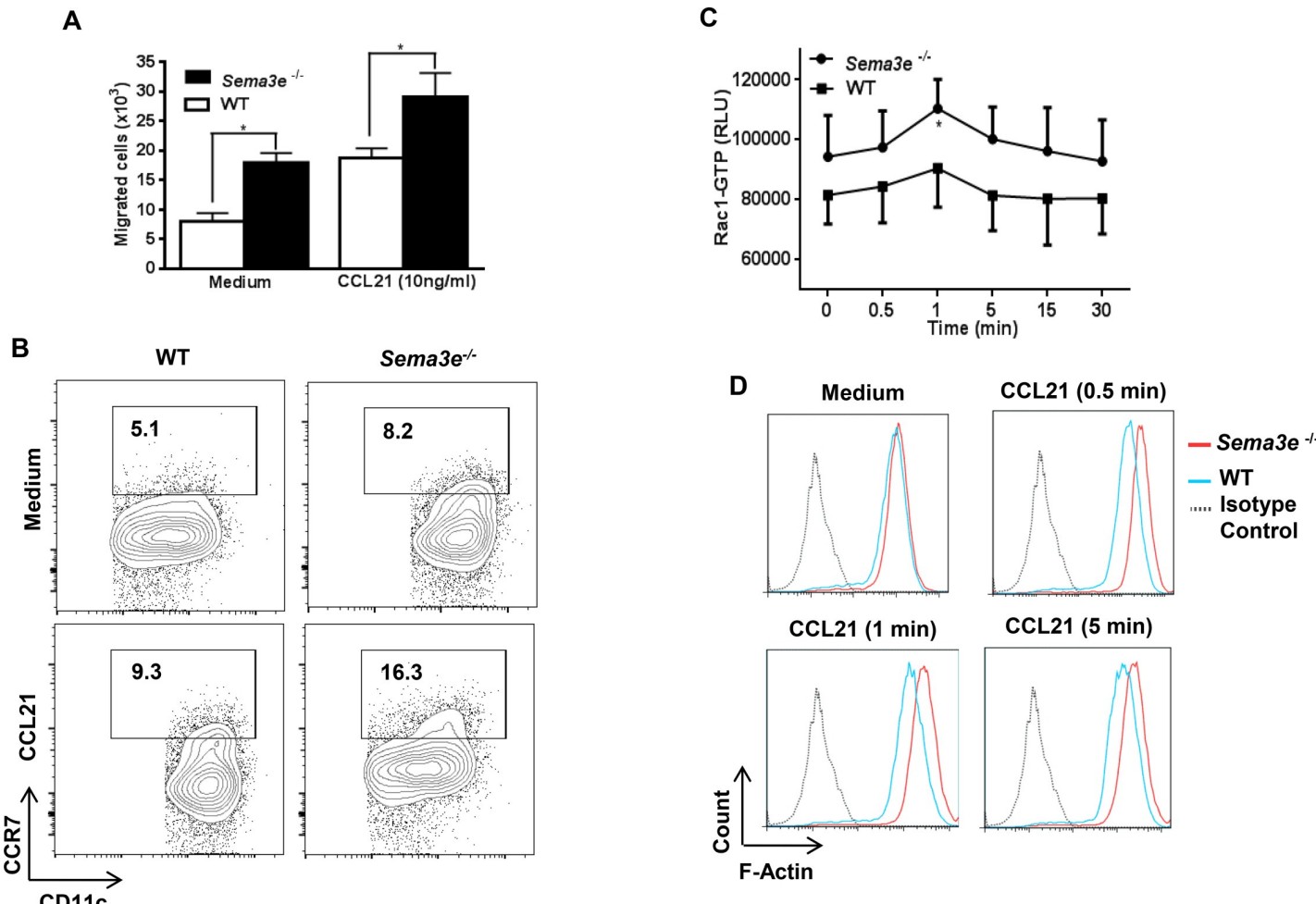

**Fig 1. Role of Sema3E in regulation of BMDC migration.** Basal and CCL21-induced migration of BMDC from *Sema3e*[-/-] and WT mice was assessed by transwell assay *in vitro* (A). Surface expression of CCR7 is increased in BMDC from *Sema3e*[-/-] mice compare to those of WT littermates as determined by flow cytometry at the baseline or after stimulation with CCL21 (B). Rac1 GTPase activity was compared between BMDC from *Sema3e*[-/-] and WT mice before and after CCL21 stimulation by performing G-LISA (C). Kinetic study of actin polymerization by Phalloidin staining at the baseline and upon stimulation with CCL21 revealed higher F-actin content in the absence of Sema3E over time (D). Data represent at least three independent experiments. (n = 3–6 per group, *P<0.05).

## Sema3E is implicated in the pulmonary migration of CD11b[+] cDC during allergen sensitization

It has been shown that pulmonary dendritic cells migrate to the MLN upon allergen encounter via CCR7 as an essential mediator of DC migration [3]. To delineate the potential impaired regulatory mechanism underlying hyper-inflammatory phenotype in the absence of Sema3E, we established an *in vivo* model of allergen sensitization by intranasal administration of a single high dose of HDM (100 μg) for 36 and 72 hours (Fig 2A) [3, 10]. First, we observed an elevation of total cDC population, CD11c[+] MHCII[hi], in MLN from *Sema3e*[-/-] mice after HDM sensitization (Fig 2B). Then, we studied the surface expression of CCR7 on cDC subsets from MLN. HDM exposure for 3 days induced higher CCR7 expression in CD11b[+] MLN cDC from *Sema3e*[-/-] compared to those of WT mice (Fig 2B and 2C) which was not significantly different earlier (36h, data not shown). On the other hand, the frequency of CD11b[+] cDC was increased in the lungs, but not MLN, 36 hours post-sensitization (Fig 2C). Collectively, our *in vivo* data

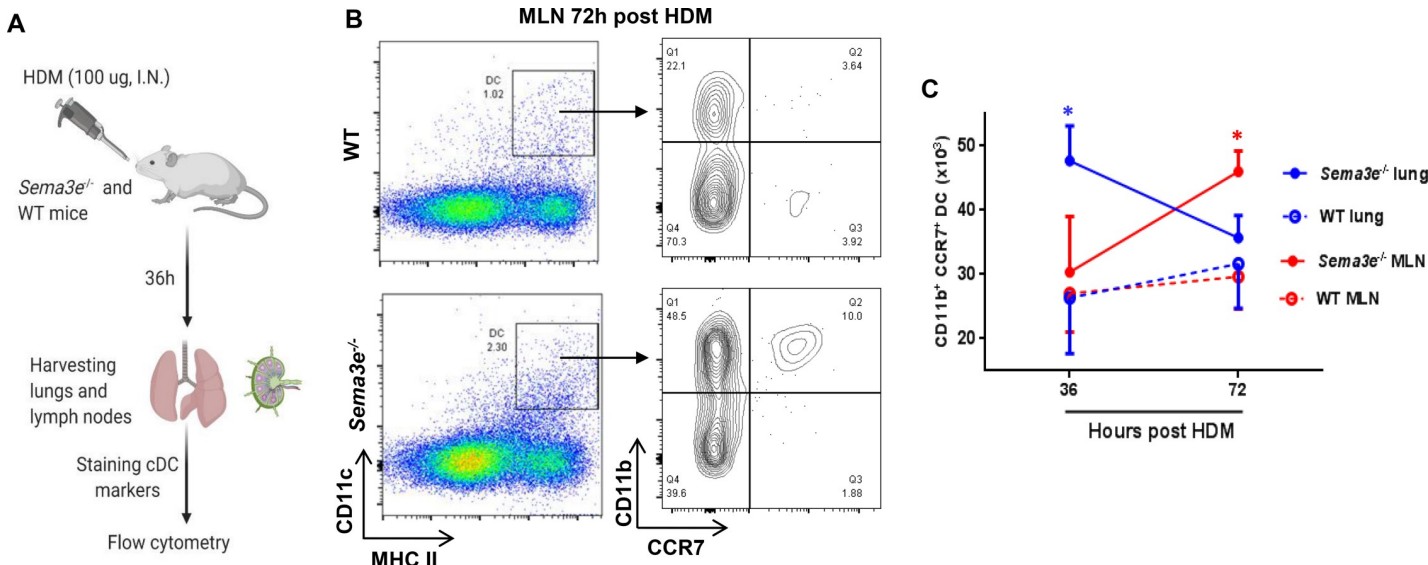

**Fig 2. HDM sensitization induces upregulation of CCR7 in *Sema3e*<sup>-/-</sup> pulmonary dendritic cells.** A schematic representation of *in vivo* model to induce HDM sensitization (A). CCR7 surface expression in CD11b<sup>+</sup> cDC from the lung and MLN was compared between *Sema3e*<sup>-/-</sup> and WT mice after 36 or 72h intranasal administration of HDM (B). Statistical comparison of CD11b<sup>+</sup> CCR7<sup>+</sup> cDC from lungs or MLN between *Sema3e*<sup>-/-</sup> and WT mice 36 and 72h post-sensitization (n = 3–6 per group, *P<0.05*).

support the notion that increased CCR7 expression in *Sema3e*<sup>-/-</sup>mice leads to enhanced accumulation of pulmonary dendritic cells particularly CD11b<sup>+</sup> cDC subset.

## Allergen-induced expression of PD-L2 and IRF-4 is increased in pulmonary CD11b<sup>+</sup> cDC from *Sema3e*<sup>-/-</sup> mice

Considering the key role of PD-L2 and IRF-4 in development of allergic type 2 inflammation via regulation of DC functions, we examined whether Sema3E could contribute to expression of these proteins in HDM sensitization model. We have demonstrated our gating strategy by which we have determined major cDC subsets (CD11b<sup>+</sup> vs CD103<sup>+</sup>) by excluding dead cells, debris, autofluorescent macrophages and then including MHCII<sup>hi</sup> CD11c<sup>+</sup> cells (S3 Fig). Our flow cytometry data revealed that surface expression of PD-L2 is not different between CD11b<sup>+</sup> cDC from *Sema3e*<sup>-/-</sup> vs WT mice at the baseline. However, HDM-induced PD-L2 surface expression was significantly more pronounced on CD11b<sup>+</sup> cDC from *Sema3e*<sup>-/-</sup> than WT control mice as shown in Fig 3A–3C. Then, utilizing the same gating strategy as that of PD-L2 (S3 Fig), we performed intranuclear staining to assess the expression IRF-4 which is an essential transcription factor in regulation of myeloid dendritic cells. As shown in Fig 3D and 3E, both frequency and number of IRF-4 expressing CD11b<sup>+</sup> cDC from *Sema3e*<sup>-/-</sup> mice (solid line) was significantly increased in the absence of Sema3E compared to WT littermates (dotted line) upon HDM sensitization. Therefore, our data indicate that genetic deletion of Sema3E is associated with not only phenotypic alteration of cDC subsets but also differential expression of a functional signature including PD-L2, IRF-4, and CCR7 in these cells.

## Allergen uptake capacity is elevated in pulmonary CD11b<sup>+</sup> cDC from *Sema3e*<sup>-/-</sup> mice during sensitization

Dendritic cells are known as the professional antigen presenting cells. Particularly, the capacity of CD11b<sup>+</sup> cDC to uptake an allergen determines the consequent presentation upon

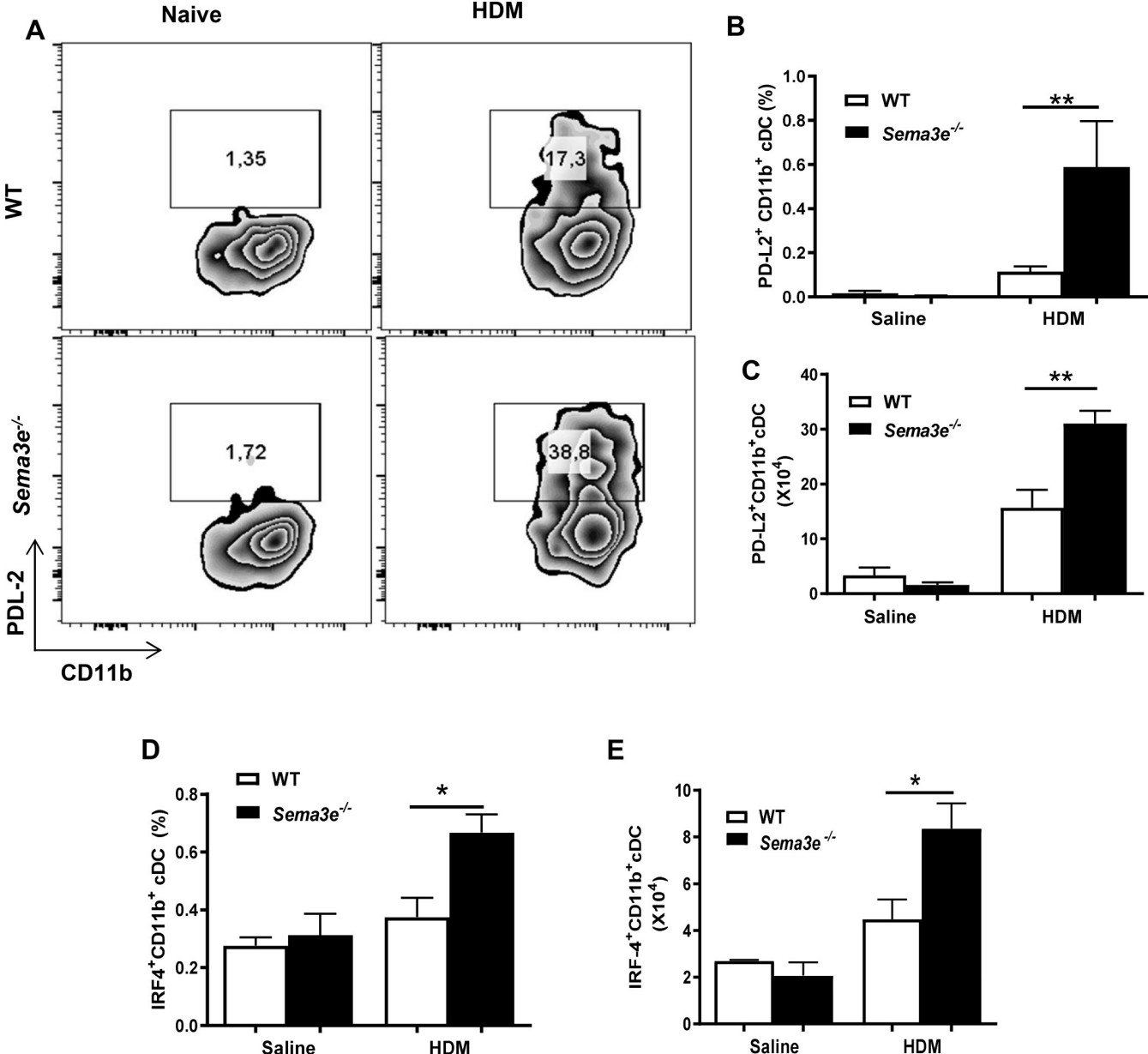

**Fig 3. Increased expression of PD-L2 and IF-4 in CD11b⁺ pulmonary cDC in *Sema3e* ⁻/⁻ mice exposed to HDM.** (A-C) Surface expression of PD-L2 was measured on C11b⁺ DC subset from the lungs using flow cytometry. Data represent at least two independent experiments (n = 3–5 mice per group, ***$P<0.001$). (D, E) IRF4 intranuclear protein expression was detected by flow cytometry. Isotype control validates the specificity of staining. Cells were pre-gated as described in Fig 3. Data represent at least two independent experiments, n = 3–5 mice per group. *$P<0.05$.

processing. Therefore, we investigated the potential role of Sema3E in antigen uptake by pulmonary CD11b⁺ cDC in our sensitization model in which HDM was conjugated with APC fluorochrome prior to intranasal administration (Fig 4A). As shown in Fig 4A, the amount of HDM uptake by pulmonary CD11b⁺ was remarkably higher than CD103⁺ cDC. Surprisingly, Sema3E deficiency led to an enhanced antigen uptake in CD11b⁺ cDC compared to WT control group 36h post-sensitization in the lungs. However, in accordance with the literature, the level of uptake was higher in CD11b⁺ than CD103⁺ cDC (Fig 4B and 4C). Collectively, these results suggest a potential role of Sema3E in the regulation of allergen uptake.

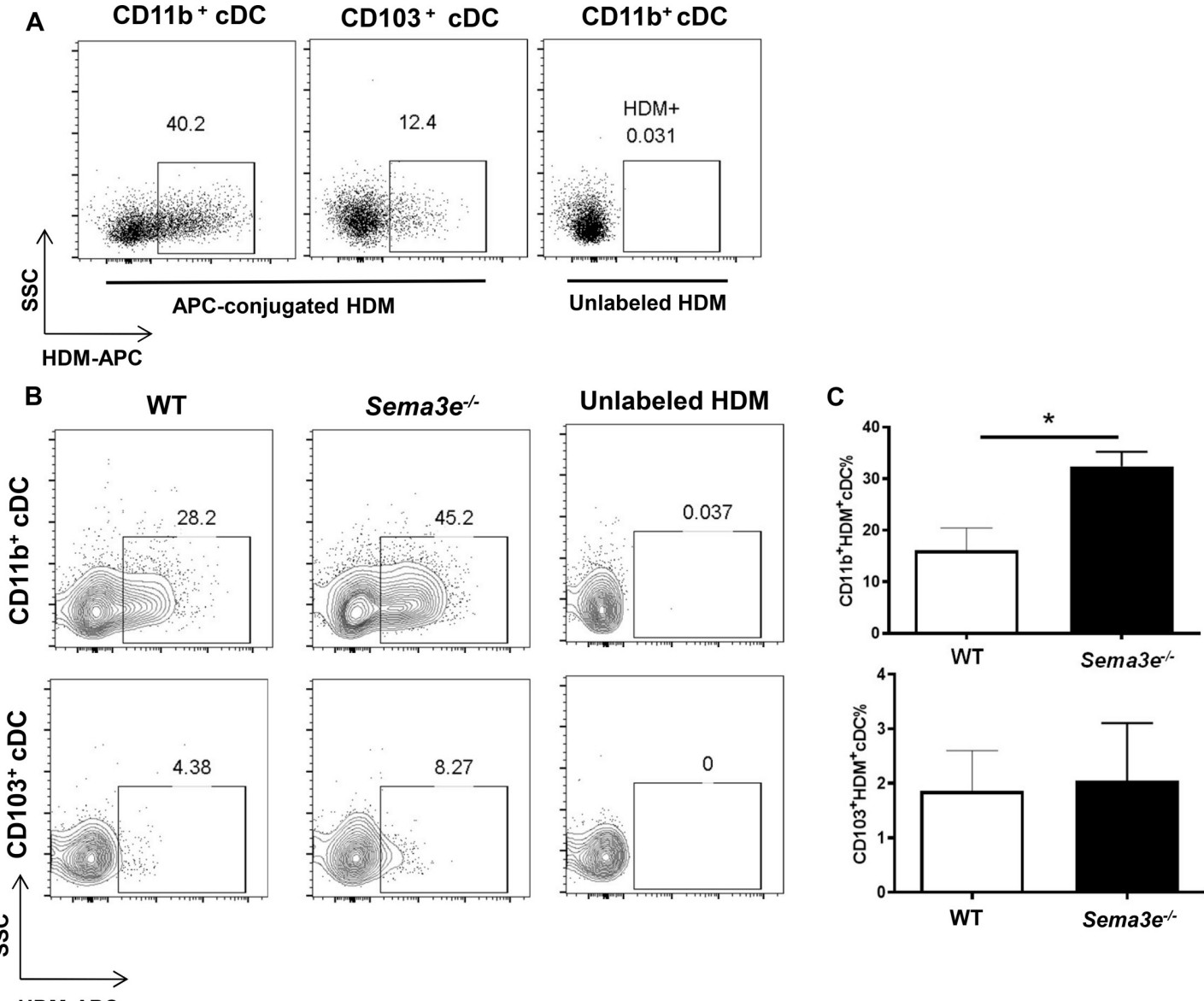

**Fig 4. Enhanced uptake of HDM in pulmonary cDC subsets from *Sema3e⁻/⁻* mice.** HDM allergen was fluorescently labelled using a conjugation kit and intranasally administered to the mice followed by acquiring the samples and phenotyping CD11b⁺ vs CD103⁺ cDC subsets (A). The level of HDM uptake by pulmonary cDC subsets was compared between *Sema3e⁻/⁻* and WT mice. cDC from the mice sensitized with unlabeled HDM was served as a gating control (n = 3 per group) (B–C).

## Discussion

In this study, we have first utilized an *in vitro* and *in vivo* approach by which the mechanism underlying the regulatory role of Sema3E in DC functions has been addressed. We have shown that BMDC from *Sema3e⁻/⁻* mice have a hyper migratory phenotype which could be mediated specifically by upregulation of CCL21-CCR7 chemotactic axis, higher activation of Rac1 signaling and F-actin polymerization than those of WT controls. Furthermore, our *in vivo* studies reveal that compared to WT, the absence of Sema3e signaling, transcription factor IRF4 and surface marker PD-L2 expression augmented in DC after treatment with a Th2/Th17 promoting signal, HDM. These events were accompanied with an enhanced ability of

antigen uptake. Taken together, our data point out to an important role of Sema3e in regulating important regulatory signal that impact DC cell function in the lung.

It has been reported that small GTPases such as RhoA, cell division control protein 42 homolog (CDC42) and Rac tightly regulate DC migration, antigen uptake and T cell priming [16–18]. Furthermore, Sema3E inhibitory effect on macrophages [19] and ASM cells [20] is mediated through Rac1 signaling pathway. *Tata et al.* have revealed that Sema3E-induces hyper-collapse of endothelial cells after silencing *Rac1* gene expression which is rescued upon treatment with constitutively active Rac1 [21]. Elevated Rac1 GTPase activity in mature BMDC from *Sema3e*[-/-] mice may explain the mechanism underlying higher migration as well as T cell priming, reported in our previous study [10], in the absence of Sema3E.

Increased F-actin content and higher surface expression of CCR7 in BMDC from *Sema3e*[-/-] mice may further provide potential mechanistic evidence behind their higher migration ability compared to those of WT littermates which was further supported by higher level of CCR7 induction in CD11b[+] pulmonary DC upon HDM exposure *in vivo*. Moreover, Rac1 is an essential signaling component downstream of CCR7 pathway that controls ERK signaling pathway activation in DC [22]. Thus, higher GTPase activity could connect CCR7 over-expression to consequent increased F-actin polymerization in *Sema3e*[-/-] BMDC upon CCL21 stimulation. Because of the low frequency of highly pure CD11b[+] pulmonary DC, performing Rac1 GTPase activity and F-actin polymerization assays were not feasible on these cells after sorting. It should be mentioned that there might be other potential signaling targets involved in regulation of Sema3E-mediated effects in DC which have remained elusive, so far. However, CCR2 surface expression was not significantly different between BMDC differentiated from *Sema3e*[-/-] and WT mice. In addition, BMDC migration was not significantly different upon stimulation with CCL2; while stimulation with CCL19, as an alternative ligand for CCR7, significantly increased migration of BMDC from *Sema3e*[-/-] mice compared to WT controls *in vitro* (data not shown).

Since BMDC have been directly differentiated from the progenitor cells at presence of GM-CSF, the potential impact of Sema3E on development of immune cells under healthy and pathological, conditions e.g. allergic asthma, was further investigated. Our *in vivo* model focuses on sensitization phase of exposure to an allergen which is crucial in the context of DC innate functions to shape the allergic response. Our previously published data on the impact of Sema3E on cDC phenotype is based on a 2-week HDM exposure model which represents a challenge phase further supported by current functional findings. We have mainly focused on CD11b[+] cDC since they have been shown to induce type 2 inflammation in allergic asthma. It requires migration of CD11b[+] cDC from the lungs to mediastinal lymph nodes where they present the aeroallergens to naïve T cells in a CCR7-dependent fashion. We have provided mechanistic evidence that CCR7 expression is kinetically upregulated in the absence of Sema3E suggestive of a novel target to modulate unwanted DC migration, e.g. in allergic or autoimmune diseases.

Overt uptake of HDM in pulmonary cDC from *Sema3e*[-/-] mice indicates that suppression of Sema3E, as we reported in severe asthmatic patients [23], could subsequently lead to an increased allergen presentation and heighten the disease immunopathology. This notion is further supported by significant increase in upregulation of PD-L2 and IRF-4 as key mediators in type 2 allergic immunity in CD11b[+] cDC from *Sema3e*[-/-] upon HDM sensitization.

Lewkowich *et al.* have previously reported that PD-L2 expression allergen exposure in mice upregulates PD-L2 expression on pulmonary DCs and its blockade decreases allergic airway hyperresponsiveness [2]. Of clinical importance, they have further demonstrated that PD-L2 expression in bronchial biopsies correlated with the severity of asthma. The pro-allergic mechanism of PD-L2 action in asthma mouse model has been suggested to be mediated by

diminishing IL-12 production in DC which counterbalancing IL-13 expression as a key type 2 cytokine involved in AHR [2]. Targeting PD-L2 using its ligand, repulsive guidance molecule b (RGMb), was proposed a therapeutic approach for allergic asthma [24]. However, it has been recently revealed that the anti-allergic effect of RGMb in a mouse model of asthma is independent of PD-L2 interaction further confirmed in PD-L2 deficient mice sensitized to OVA which may require to be validated in protease allergen models such as HDM [25]. In addition, Minkyoung e*t al*. have reported a complex mechanism in which protease-fibrinogen cleavage products-TLR4-mast cell-IL-13 axis favors development of pro-allergic/asthmatic PD-L2[+] DCs in mice [26]. Finally, in vitro administration of blocking antibody against PD-L2 remarkably inhibits IL-5 and IL-13 production by myeloid DCs obtained from patients with persistent asthma [27]; a therapeutic option which could be achieved by replenishing Sema3E.

Similar to PD-L2, expression of IRF-4 is increased in patients with allergic asthma [28]. IRF4- expressing DC are important for the DC-driven polarization of Th17 responses in the intestine and lung, for the induction of Th2 responses via CD11c[+] DC in lung allergy and skin parasite models, and for attenuation of Th1 responses [4, 5]. Also, inhibition of IRF4 in cDC block type 2 inflammation while skewing the immune system towards a Th17 response [4] which could be detrimental for Th2-low phenotype in asthmatic patients. This effect has been showed to be independent and dependent of engaging pattern recognition receptors. However, our results represent Sema3E as an essential regulator of IRF4 expression with counterbalancing effects on both Th2 and Th17 responses.

Future studies will determine whether replenishment of Sema3E by administering exogenous recombinant Sema3E or its peptide derivatives during or before sensitization would inhibit dysregulated HDM uptake as well as overexpression of CCR7, PD-L2, IRF-4 in CD11b[+] cDC.

Altogether, this study provides novel mechanistic insights to explain the key role of Sema3E in DC biology in general and pro-allergic function of CD11b[+] cDC in particular which had not been previously addressed. Combined with our previous reports on the regulatory role of Sema3E in allergic inflammation, neutrophil migration, and also airway smooth muscle cell function, targeting this guidance cue may be considered a comprehensive approach in immunopathological disorders in which immune regulation is impaired.

## Supporting information

**S1 Fig. *In vitro* GM-CSF-mediated differentiation of BMDC as confirmed by measuring CD11c expression (Related to Fig 1).**
(TIF)

**S2 Fig. Enhanced HDM-induced CCR7 expression on BMDC from *Sema3e*[-/-] mice (Related to Fig 1).**
(TIF)

**S3 Fig. Gating strategy in flow cytometry experiments to determine pulmonary cDC subsets.** In total cell population (R1), single (R2) live (R3) cells were selected and macrophages were excluded (R4). Then, total pulmonary cDC were determined based on high surface expression of MHCII and positivity for CD11c (R5). Finally, distinct CD11b vs CD103 expressing cDC were characterized (R6).
(TIF)

## Acknowledgments

The authors would like to thank Dr. Fanny Mann (Developmental Biology Institute of Marseille Luminy, Université de la Méditerranée, Marseille, France) for providing us with

*Sema3e$^{-/-}$* mouse model and Dr. Christine Zhang (Flow cytometry core facility, University of Manitoba, Winnipeg, Canada) for her help on sorting lung dendritic cells.

## Author Contributions

**Conceptualization:** Hesam Movassagh, Sam K. P. Kung, Abdelilah S. Gounni.

**Data curation:** Hesam Movassagh, Lianyu Shan, Nazila Ariaee.

**Formal analysis:** Hesam Movassagh, Abdulaziz Alamri, Nazila Ariaee, Sam K. P. Kung, Abdelilah S. Gounni.

**Investigation:** Abdelilah S. Gounni.

**Methodology:** Hesam Movassagh, Lianyu Shan, Latifa Koussih, Abdulaziz Alamri.

**Project administration:** Lianyu Shan, Latifa Koussih, Abdelilah S. Gounni.

**Resources:** Abdelilah S. Gounni.

**Supervision:** Abdelilah S. Gounni.

**Validation:** Abdelilah S. Gounni.

**Writing – original draft:** Hesam Movassagh.

**Writing – review & editing:** Latifa Koussih, Abdulaziz Alamri, Nazila Ariaee, Sam K. P. Kung, Abdelilah S. Gounni.

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
