## [Decision Letter · Decision Letter 0]

26 Feb 2021

PONE-D-21-02602

Semaphorin 3E Deficiency Dysregulates Dendritic Cell Functions: In Vitro and In Vivo Evidence

PLOS ONE

Dear Dr. Gounni,

Thank you for submitting your manuscript to PLOS ONE. After careful consideration, we feel that it has merit but does not fully meet PLOS ONE’s publication criteria as it currently stands. Therefore, we invite you to submit a revised version of the manuscript that addresses the points raised during the review process.

We look forward to receiving your revised manuscript.

Kind regards,

Amarjit Mishra, PhD

Academic Editor

PLOS ONE

Journal Requirements:

Reviewers' comments:

Reviewer's Responses to Questions

**Comments to the Author**

1. Is the manuscript technically sound, and do the data support the conclusions?

Reviewer #1: Partly

Reviewer #2: Yes

2. Has the statistical analysis been performed appropriately and rigorously? 

Reviewer #1: Yes

Reviewer #2: Yes

3. Have the authors made all data underlying the findings in their manuscript fully available?

Reviewer #1: Yes

Reviewer #2: Yes

4. Is the manuscript presented in an intelligible fashion and written in standard English?

Reviewer #1: Yes

Reviewer #2: Yes

5. Review Comments to the Author

Reviewer #1: Comments for the Author:

In the present manuscript authors intends to elucidate the non-redundant role of Semaphorin 3E (Sema3E) in Dendritic cell migration and antigen uptake during allergen sensitization in mice. They have shown increased migratory phenotype of Sema3E deficient BMDCs with increased CCR7 expression compared to Sema3E sufficient BMDCs after culture with ligand CCL21. Further, author have shown increased Rac1 GTPase activity and F-actin expression in absence of Sema3E. Further, author have indicated increased accumulation of lung CD11b+ cDCs in Sema3E knock out mice and enhanced expression of PD-L2 and IRF-4 compared to wild type controls.

Author have tried to elucidate functional importance Sema3E in the inflammation, specifically during sensitization phase which is very interesting and relevant to the field. However, there are few concerns regarding experiments, figures and data which need to be answered before publication of this manuscript. My specific comments are:

Comments:

1. Author have used in vitro BMDCs model to show the migratory defect in Sema3E deficient BMDCs compared to Sema3E sufficient one. The data showing increased CCR7 expression in Sema3E deficient BMDCs in presence of CCL21 looks promising. However, flow data presented is not convincing as basal CD11c expression in Sema3E deficient BMDCs seems high while after adding CCL21 it comes to normal. Author should also include the data with alternative ligand CCL19 which could add to conclusion that CCL21 and CCL19 both lead to the CCR7 pathway activation controlling the migration of DCs.

2. After sensitization with HDM-APC it is clearly shown the increased uptake of allergen by lung CD11b+cDCs of Sema3E deficient mice. It would better idea to add the HDM-APC+ CD11b+cDCs data from mediastinal lymph node with same experiment. Also, it is not mentioned how many mice were used to perform this experiment.

3. Figure 3, the representative flow plot for gating strategy do not add to results, it could be moved to supplemental data. What is FITC in SSC vs FITC? Is that representing SiglecF?

4. Increased expression level of PD-L2 and IRF-4 in lung cDCs seems very promising. What are the numbers representing? Is the data from a whole lung?

5. Apart from CCR7, what was the expression level of other activation markers in the DCs during the sensitization phase? Did author check for activation status of maturation markers in the DCs from Sema3E deficient mice?

6. Figure1B, flow plots are not uniform that needs to be replaced.

7. Figure2A, HDM amount (100 ug) contradicts the materials methods (page7, line 126, 1 ug).

8. Page7, line 116, clone 418 should be N418.

9. Page2: Line 29, replace ‘time’ with ‘type’.

10. Page5: Line 80, replace ‘BMDC’ with ‘BMDCs’.

11. Page14: Line 273, OF should be corrected.

Reviewer #2: In this study, Gounni et al, have explored the role of semaphoring 3E in dendritic cell differentiation and migration along with, the contributory role of CCR7 expression, elevated rac1 GTPase activity, and F-actin polymerization. The authors have successfully demonstrated that, Sema3E deficiency leads to an enhanced expression of PD-L2 and IRF-4, which results in enhanced antigen uptake by differentiated dendritic cells.

This research has merit as it will help in development of novel immunotherapeutic strategies for the treatment of allergen based diseases in which immune regulation is impaired.

Authors, have presented the data clearly and have conducted all the necessary assays and statistical analysis.

6. PLOS authors have the option to publish the peer review history of their article (what does this mean?). If published, this will include your full peer review and any attached files.

Reviewer #1: **Yes: **Anil Kumar Jaiswal

Reviewer #2: No

---

## [Author Response · Author response to Decision Letter 0]

11 May 2021

Reviewer #1)

1. Author have used in vitro BMDCs model to show the migratory defect in Sema3E deficient BMDCs compared to Sema3E sufficient one. The data showing increased CCR7 expression in Sema3E deficient BMDCs in presence of CCL21 looks promising. However, flow data presented is not convincing as basal CD11c expression in Sema3E deficient BMDCs seems high while after adding CCL21 it comes to normal. Author should also include the data with alternative ligand CCL19 which could add to conclusion that CCL21 and CCL19 both lead to the CCR7 pathway activation controlling the migration of DCs.

Reply: Thank you for your comment. As presented in Fig. 1B, the percentage of CCR7 expressing CD11c+ cells from WT mice have been increased from 5% at the baseline to 9% after stimulation with CCL21. As a novel finding we have shown that genetic deletion of Sema3E increased the baseline expression of CCR7 which explains why BMDC from KO mice have an increased basal migration (Fig.1A). This phenotype sustained after CCL21 stimulation. We have observed that stimulation of BMDC from Sema3E KO mice with CCL19 induces cell migration more robustly than those of WT controls. Unfortunately, we have not investigated whether CCL19 would similarly signal as of CCL21. 

2. After sensitization with HDM-APC it is clearly shown the increased uptake of allergen by lung CD11b+cDCs of Sema3E deficient mice. It would better idea to add the HDM-APC+ CD11b+cDCs data from mediastinal lymph node with same experiment. Also, it is not mentioned how many mice were used to perform this experiment.

Reply: Thank you for your comment. There is a technical issue regarding the low number of this specific cell subset in the MLN post-sensitization. In fact, the size of lymph nodes is very small 3 days after exposure to one dose of HDM which makes the FACS isolation of these cells (pre-gated as explained in the manuscript) quite challenging. 

3. Figure 3, the representative flow plot for gating strategy do not add to results, it could be moved to supplemental data. What is FITC in SSC vs FITC? Is that representing SiglecF?

Reply: Thank you for your comment. We have removed the gating strategy and presented it as a supplemental figure in the revised manuscript. FITC is a dumb channel to exclude autofluorescent macrophages. 

4. Increased expression level of PD-L2 and IRF-4 in lung cDCs seems very promising. What are the numbers representing? Is the data from a whole lung?

Reply: Thank you for your comment. The numbers represent the absolute counts of corresponding cell subsets from the total live whole lung cells. 

5. Apart from CCR7, what was the expression level of other activation markers in the DCs during the sensitization phase? Did author check for activation status of maturation markers in the DCs from Sema3E deficient mice?

Reply: Thank you for your comment. As a hypothesis-driven approach to address the mechanism underlying increased DC migration in the absence of Sema3E, we specifically focused on CCR7 as an essential receptor in migration of dendritic cells and homing in the lungs. 

6. Figure1B, flow plots are not uniform that needs to be replaced.

Reply: Thank you for your comment. We double checked the flow plots in Figure 1B. We think the main reason they do not look uniform is the impact of Sema3E deletion as well as stimulation with CCL21 which directly affect shifting the Y-axis upward indicating upregulation of CCR7. 

7. Figure2A, HDM amount (100 ug) contradicts the materials methods (page7, line 126, 1 ug). 

8. Page7, line 116, clone 418 should be N418.

9. Page2: Line 29, replace ‘time’ with ‘type’.

10. Page5: Line 80, replace ‘BMDC’ with ‘BMDCs’.

11. Page14: Line 273, OF should be corrected in the revised version.

Reply: Sorry for the typos. We have corrected all of them in the revised version as highlighted in yellow.

Reviewer #2)

 This research has merit as it will help in development of novel immunotherapeutic strategies for the treatment of allergen based diseases in which immune regulation is impaired. Authors have presented the data clearly and have conducted all the necessary assays and statistical analysis.

Reply: Thank you very much.

---

## [Decision Letter · Decision Letter 1]

25 May 2021

Semaphorin 3E Deficiency Dysregulates Dendritic Cell Functions: In Vitro and In Vivo Evidence

PONE-D-21-02602R1

Dear Dr. Gounni,

We’re pleased to inform you that your manuscript has been judged scientifically suitable for publication and will be formally accepted for publication once it meets all outstanding technical requirements.

Kind regards,

Amarjit Mishra, PhD

Academic Editor

PLOS ONE

Reviewers' comments:

Reviewer's Responses to Questions

**Comments to the Author**

1. If the authors have adequately addressed your comments raised in a previous round of review and you feel that this manuscript is now acceptable for publication, you may indicate that here to bypass the “Comments to the Author” section, enter your conflict of interest statement in the “Confidential to Editor” section, and submit your "Accept" recommendation.

Reviewer #1: All comments have been addressed

2. Is the manuscript technically sound, and do the data support the conclusions?

Reviewer #1: Partly

3. Has the statistical analysis been performed appropriately and rigorously? 

Reviewer #1: Yes

4. Have the authors made all data underlying the findings in their manuscript fully available?

Reviewer #1: Yes

5. Is the manuscript presented in an intelligible fashion and written in standard English?

Reviewer #1: Yes

6. Review Comments to the Author

Reviewer #1: Comments for the Author:

In the present manuscript Movassagh et. al intends to show role of Semaphorin 3E (Sema3E) in Dendritic cell migration and antigen uptake during allergen sensitization in mice. During, first review author has address most of the issue. However, few technical issues remain the same.

Comments:

1. Figure1B: This figure is not impressive, and author failed to address the question. The flow contour plot shows that addition of CCL21 to Sema3e-/- BMDC leads to increased number of CD11c-low expressing cells while it is reverse with WT BMDCs.

7. PLOS authors have the option to publish the peer review history of their article (what does this mean?). If published, this will include your full peer review and any attached files.

Reviewer #1: **Yes: **Anil Kumar Jaiswal

---

## [Editor Report · Acceptance letter]

7 Jun 2021

PONE-D-21-02602R1 

Semaphorin 3E Deficiency Dysregulates Dendritic Cell Functions: In Vitro and In Vivo Evidence 

Dear Dr. Gounni:

I'm pleased to inform you that your manuscript has been deemed suitable for publication in PLOS ONE. Congratulations! Your manuscript is now with our production department. 

Kind regards, 

on behalf of

Dr. Amarjit Mishra 

Academic Editor

PLOS ONE